# Rapid Visual Detection of High Nitrogen-Use Efficiency Gene *OsGRF4* in Rice (*Oryza sativa* L.) Using Loop-Mediated Isothermal Amplification Method

**DOI:** 10.3390/genes14101850

**Published:** 2023-09-23

**Authors:** Yonghang Tian, Wenwei Ye, Xiangshuai Liang, Peizhou Xu, Xianjun Wu, Xiangdong Fu, Yaoxian Chin, Yongxiang Liao

**Affiliations:** 1College of Food Science and Engineering, Hainan Tropical Ocean University, No. 1 Yucai Road, Sanya 572022, China; isxiaoshuai@outlook.com (X.L.); chinyx1@163.com (Y.C.); 2Marine Food Engineering Technology Research Center of Hainan Province, No. 1 Yucai Road, Sanya 572022, China; 3Rice Research Institute, Sichuan Agricultural University, No. 211 Huiming Road, Wenjiang District, Chengdu 611130, China; yewenwei0723@163.com (W.Y.); xpzhxj@163.com (P.X.); wuxjsau@126.com (X.W.); 4Institute of Genetics and Developmental Biology, Chinese Academy of Sciences, No. 1 West Beichen Road, Chaoyang District, Beijing 100101, China; xdfu@genetics.ac.cn

**Keywords:** rice, NUE, *OsGRF4*, CAPs, LAMP, HNB

## Abstract

The *GROWTH-REGULATING FACTOR4* (*OsGRF4)* allele is an important target for the development of new high nitrogen-use efficiency (NUE) rice lines that would require less fertilizers. Detection of *OsGRF4* through PCR (polymerase chain reaction)-based assay is cumbersome and needs advanced laboratory skills and facilities. Hence, a method for conveniently and rapidly detecting *OsGRF4* on-field is a key requirement for further research and applications. In this study, we employed cleaved amplified polymorphic sequences (CAPs) and loop-mediated isothermal amplification (LAMP) techniques to develop a convenient visual detection method for high NUE gene *OsGRF4*^NM73^ (*OsGRF4* from the rice line NM73). The TC→AA mutation at 1187–1188 bp loci was selected as the target sequence for the *OsGRF4*^NM73^ allele. We further employed this method of identification in 10 rice varieties that carried the *OsGRF4* gene and results revealed that one variety (NM73) carries the target *OsGRF4*^NM73^ allele, while other varieties did not possess the *osgrf4* genotype. The optimal LAMP reaction using hydroxynaphthol blue (HNB), a chromogenic indicator, was carried out at 65 °C for 60 min, and the presence of *OsGRF4*^NM73^ allele was confirmed by color changes from violet to sky blue. The results of this study showed that the LAMP method can be conveniently and accurately used to detect the *OsGRF4*^NM73^ gene in rice.

## 1. Introduction

In the view of global climate change coupled with an ever-growing population, sustainable cultivation of rice, a staple food for more than half of the world’s population, remains an important research area in agriculture. With growing understanding of the genetics of rice, crop yield enhancement via genetic manipulation has been studied extensively, leading to the discovery of various genes regulating desirable traits in rice. *OsGRF4*, mapped on chromosome 2 of rice, is a key gene that is extensively targeted in developing new and high-yielding rice varieties [1]. Using quantitative trait locus (QTL) analysis and map-based cloning, we previously established that *OsGRF4* is a key transcription factor that affects nitrogen use efficiency in rice [2]. As a dominant gene, *OsGRF4* could be used to enhance the grain size and yield in rice [3,4,5]. A 2 bp substitution of TC→AA at position 1187–1188 bp of exon 3 would inhibit posttranscriptional regulation by miR396c, leading to increased *OsGRF4* expression and a semi-dominant phenotype in the host plant [2]. Furthermore, the *OsGRF4* gene also regulates the transcriptional factor MYB61 to promote nitrogen utilization and biomass production in rice [6], which is crucial to the development of the sustainable farming of rice. Additionally, studies have shown that *OsGRF4* influences the physiology of the rice as well, including improving the panicle length and reducing seed shattering, which is beneficial for high-yield breeding and mechanical harvesting [7], and enhancing cold tolerance and blast resistance in rice [8,9]. However, since overexpression of *OsGRF4* is a rare occurrence in nature [3], its potential for mass production application needs to be explored further, making the accurate and convenient detection of rice germplasm carrying *OsGRF4* a vital requirement towards commercial utilization.

InDel (insertion and deletion), CAPs (cleaved amplified polymorphic sequences) and dCAPs (derived cleaved amplified polymorphic sequences) are the most popular molecular markers for the current basic research and breeding applications, and three methods are technically very stable and easy to use. If the deletion and insertion variants are larger than 20 bp, they can usually be detected by using the InDel markers in combination with agarose gel electrophoresis [10]. If there is an insertion or deletion larger than 4 bp, it can be detected directly by PAGE (polyacrylamide gel electrophoresis) [11], but the procedure for PAGE is more laborious, expensive, and time consuming than agarose gel electrophoresis. If the InDel marker cannot be used, the CAPs marker is usually preferred. CAPs is an extension of the restriction fragment length polymorphism (RFLP) method, which analyses PCR amplicons by restriction enzyme digestion [12]. The combinations of primers and restriction endonucleases are very versatile, increasing the opportunities to identify polymorphisms, are easy to use, and the digested products can be detected by agarose gel electrophoresis. In eukaryotes, CAPs markers are co-dominant and distinguish between heterozygous and homozygous genotypes. The CAPs marker method requires a small amount of sample DNA and is not stringent regarding DNA concentration. The CAPs marker method is also simple, fast, and highly automated. In the absence of suitable enzyme digestion sites for the CAPs marker method, an appropriate number of mutations can be introduced to use the dCAPs marker method. The expensive restriction endonucleases are used in the CAPs and dCAPs marker methods, which are usually much more expensive to use than the InDel marker method. In addition to InDel, CAPs, and dCAPs, there are RFLP, AFLP (amplified fragment length polymorphism), SCAR (sequence characterized amplified region), SSR (simple sequence repeats), RAPD (random amplified polymorphic DNA), ISSR (inter simple sequence repeats), TRAP (target region amplification polymorphism), ARMS (amplification-refractory mutation system PCR), KASP (kompetitive allele-specific PCR) and HRM (high-resolution melting), etc., marker methods that are more complicated to use, unstable, more costly, and inconvenient for breeding and field detection.

Presently, the detection of *OsGRF4* is mostly PCR-based, such as the development of molecular markers PT2 [13], GS2_SNP [14], and NGR2 5U S+*Tag* II [15]. However, PCR-based methods are costly and laborious, require DNA extraction, PCR amplification, restriction enzyme digestion, detection, and visualization of amplification products. The process is also restricted by instrument availability, which is inconvenient for crop breeding on-field [16]. Breeding is an art and a science, so experienced frontline breeders have used their years of field experience to select elite individuals, build populations, and test cross-combinations. However, accurate selection of individual elite plants and efficient parental improvement are difficult with experience alone, and breeding parents de novo by crossing with distantly related genetic resources will be even more time-consuming and arduous. To develop new excellent varieties quickly, breeders are more willing to make small changes to the traits of the backbone parents, which ultimately leads to a narrow genetic base of breeding resources and makes it difficult for breakthrough varieties to appear. Frontline breeders often do not use time-consuming, costly, and tedious genotyping methods, so there is now a gap between functional marker development and actual breeding [17]. Cheap, accurate, efficient, and convenient genotyping methods or services are therefore urgently needed.

LAMP is a cost-effective molecular identification technology [18,19] that provides a viable alternative to current PCR technology. The LAMP process is relatively straightforward, being able to rapidly amplify DNA with high specificity and selectivity under isothermal conditions at considerably lower temperatures than conventional PCR [18]. By adding chromogenic reagent, LAMP results can be instantly visualized with the naked eye, making it an attractive option for in situ detection. The LAMP reaction uses four primers targeting six regions of the gene and amplifies 10^9^–10^10^ copies of target sequences in 30–60 min with high speed, high efficiency, and high specificity [18,19]. To further speed up the LAMP reaction, two additional loop primers can be added to the original four primers [20]. Adding the loop primers could reduce the reaction time by almost half and greatly increases sensitivity. The results of the LAMP-coupled colorimetric reaction are visible to the naked eye [21]. The LAMP amplification reaction can be carried out at a constant temperature, which is easy to operate and does not require any special equipment. Once developed, LAMP diagnostic tools can be easily learned and performed by field workers with no prior molecular experience or laboratory infrastructure [22]. However, a disadvantage of LAMP is its sensitivity to cross-contamination, particularly material present in the aerosol [23]. Contamination can be a barrier to the progress of laboratory development of the LAMP method. During the field test, the operator should be aware of the risk of sample contamination and should attempt to avoid contamination. Currently, LAMP make up more than 60% of the molecular diagnostic kits available in the market, and has been widely used in the identification of viruses including SARS (severe acute respiratory syndrome), HIV (acquiredimmune deficiency syndrome), and 2019-nCoV (2019 novel coronavirus), bacteria and fungi, as well as medical diagnosis and food safety [24,25]. In recent years, LAMP has also been successfully implemented in fast testing of genetic modifications [26], blight-resistant genes [27], *Rhizoctonia solani* [28], *Magnaporthe oryzae* [29] and *Sarocladium oryzae* [29] in rice.

In this study, we developed a visualized detection method for the *OsGRF4* genotype in rice using CAPs [30] and LAMP markers. After genotyping the rice varieties carrying *OsGRF4* allele with the CAPs markers, the presence of the allele in different rice varieties was visually detected and validated by the LAMP method using HNB as an indicator. By developing a simple yet effective visual detection method based on LAMP markers, we would enable a quick identification system for the rice carrying the allele *OsGRF4*, providing strong technical support for the breeding of high NUE rice lines, while also aiding the development and application of visual markers for other functional genes.

## 2. Materials and Methods

### 2.1. Plant Material and Genomic DNA Isolation

The plant material consists of 10 rice varieties: Nipponbare, WYJ7, R498, ZH11, 9311, NM73, 02428, YX1B, II-32B, and ZH11-SPL14, all of which were supplied by the Rice Research Institute of Sichuan Agricultural University, Chengdu, China. Among the 10 varieties, only NM73 contains a 2 bp substitution of TC→AA in its allele *OsGRF4*^NM73^. All 10 rice varieties were cultivated in Lingshui County, Hainan province, China.

The rice genomic DNA was extracted from the leaves using the standard CTAB (hexadecylcetyltrimethylammonium bromide) method [31]. Fresh rice leaves (0.2 g) were placed in a 2 mL microcentrifuge tube and ground to a powder with the aid of liquid nitrogen. After that, 700 µL of CTAB buffer, which was preheated to 65 °C, was added into the 2 mL tube and incubated for 30 min at 65 °C. Equal volumes of a mixture of phenol, chloroform, and isopentyl alcohol were then added to extract the DNA. The precipitated DNA by an equal volume of anhydrous ethanol that was precooled to −20 °C, was rinsed with 70% ethanol and air dried. Finally, 200 µL ddH_2_O was added to dissolve the DNA, and the dissolved DNA could then be used for the PCR amplification. Due to the extremely low detection limit of the LAMP reaction, where several copies of DNA can be detected, the concentration of dissolved DNA above is sufficient to meet the requirements of the LAMP reaction. The extracted DNA was stored at −20 °C until use.

### 2.2. The Primer Designing

The nucleic acid sequence for the *OsGRF4* gene in rice was downloaded from the NCBI-nr database (https://www.ncbi.nlm.nih.gov/, accessed on 11 August 2021), the GeneBank ID of *OsGRF4* gene is BK004859.1. After the gene sequence was compared and analyzed using BLAST (https://blast.ncbi.nlm.nih.gov/Blast.cgi, accessed on 2 September 2023), the TC→AA base substitution [2] at 1187–1188 bp loci in exon 3 was selected as the target sequence along with both upstream and downstream conserved regions and identified as *OsGRF4*^NM73^. CAPs markers were then designed using Primer3 [32], while both the locations and sizes of the restriction sites on the target sequence were analyzed using NEBcutter V2.0 [33]. The LAMP primers were designed using PrimerExplorer V5 (http://primerexplorer.jp/lampv5e/index.html, accessed on 2 September 2023), with the 3′ terminal of one of the primers (F1C-F2) ending with AA bases to be specifically matched against *OsGRF4*^NM73^ gene sequence at loci 1187–1188 bp. The primers were synthesized by Sangon Biotech (Shanghai) Co., Ltd. (Shanghai, China)

### 2.3. PCR Amplification and Restriction Enzyme Digestion

The PCR amplification (KT201, TIANGEN Biotech (Beijing) Co., Ltd., Beijing, China) and restriction enzyme digestion (New England Biolabs (Beijing) LTD., Beijing, China) were performed using commercial kits according to the manufacturer’s instructions. The PCR reaction was carried out in a volume of 20 µL containing 2 µL of each primer (*OsGRF4*_2F and *OsGRF4*_2R) with 10 µM, 4 µL of 2 × Taq PCR mix, 10 µL of ddH_2_O, and 2 µL of DNA templates. PCR amplification reactions were performed on a Bio-Rad T100™ Thermal Cycler (Bio-Rad, USA). The entire PCR amplification was performed with 30 cycles of predenaturation (94 °C for 3 min), denaturation (94 °C for 30 s), annealing (suitable temperature for 30 s), extension (72 °C for 1 min), and final extension (72 °C for 5 min). The restriction enzyme reaction was carried out in a volume of 10 µL containing 1 µL of 2 × CutSmart Buffer, 4 µL of PCR amplification products, 0.2 µL of Hpy188III restriction endonuclease, and 4.8 µL of ddH_2_O. The restriction enzyme reactions were performed on a TGrade Lite heating block (OSE-DB-05/06, TIANGEN Biotech (Beijing) Co., Ltd., Beijing, China). The restriction enzyme reaction was performed with restriction enzyme cleavage (37 °C for 2.5 h) and enzyme inactivation (85 °C for 20 min). Both PCR and digested products were verified by 3% agarose electrophoresis (Labnet Enduro^TM^ GDST gel imaging system, Labnet, Edison, NJ, USA).

### 2.4. LAMP Amplification

The LAMP amplification reaction was conducted using a commercial kit according to the manufacturer’s instructions (2 × Lamp PCR Master Mix (Universal), Sangon Biotech (Shanghai) Co., Ltd., Shanghai, China) at a 12.5 µL reaction volume: 6.25 µL of 2 × LAMP Mix Buffer, 10 µM F3/B3 each 1 µL, 10 µM FIP/BIP each 0.25 µL, DNA template 0.5 µL, 0.16 U/µL DNA polymerase 0.25 µL. The reaction procedure was conducted according to the literature [18] with minor adjustments. Briefly, the sample was incubated at 65° C for 1 h before inactivation of enzyme activity at 80 °C for 10 min, followed by reaction termination at 12 °C for 5 min.

The LAMP amplification reaction was performed on a TGrade Lite heating block (OSE-DB-05/06, TIANGEN Biotech (Beijing) Co., Ltd., Beijing, China). The amplification products were detected in a 3% agarose gel with the Labnet Enduro^TM^ GDST gel imaging system (Labnet, Edison, NJ, USA).

### 2.5. Visual Detection of LAMP Reaction

The LAMP colorimetric reaction was performed according to the instructions of a commercial kit (Bst 2.0 HNB Amplification Kit, A3802, HaiGene Biotech Co., Ltd., Harbin, China) at a 12.5 µL reaction volume: 2 × HNB IsoAmp Mix 6.25 µL (including HNB), primers 2.5 µL (FIP/BIP 16 µM, F3/B3 2 µM), DNA template 1 µL, ddH_2_O 2.25 µL, DNA polymerase 0.5 µL. The amplification protocol was the same as described in 2.4. Upon the end of the reaction, the presence of the *OsGRF4*^NM73^ allele was confirmed by visual inspection for color change from violet to sky blue [34].

## 3. Results

### 3.1. Designing of CAPs and LAMP Primer Sets

The CAPs primer pairs (*OsGRF4*_2F and *OsGRF4*_2R) for all the rice lines carrying the *OsGRF4*^NM73^ gene were designed using Primer3 (Figure 1, Table 1). The amplicon size was 459 bp and contains the sequence “TCAAGA” which acts as the restriction site for endonuclease Hpy188III to produce a 279 bp and 180 bp fragment, respectively, upon digestion. In contrast, the equivalent location on *OsGRF4*^NM73^ allele in NM73 rice contains the mutated sequence “AAAAGA” instead, and would thus be indigestible by Hpy188III.

The LAMP primers F3, B3, FIP, and FIP (Table 1) for the *OsGRF4*^NM73^ gene were calculated with PrimerExplorer V5 to specifically match six conserved regions in the gene. (Figure 1). Except for a TC→AA substitution at gene loci 1187–1188 bp, these segments are consistent across over 200 known rice genomes, indicating the highly specific nature of the primers. As any mismatches at the 3′ end of the primer would affect the LAMP amplification [35], the 3′ end for F2 of the primer FIP was designed to end with the AA bases at the locations 1187–1188 bp, precisely matching the *OsGRF4*^NM73^ sequence, and thus preventing amplification when the bases were replaced with TC in the wild type.

### 3.2. Optimization of PCR Reaction Conditions

To guarantee reliable digestion results, we first optimized the annealing temperature of the PCR reaction (Figure 2). The DNA of Nipponbare was used as DNA templates. Electrophoresis results showed that our primers generated a distinct, single band with an amplicon size of 400–500 bp at all tested annealing temperatures between 56.0–65.0 °C. Hence, the standard annealing temperature of 56.0 °C was adopted for subsequent amplifications.

According to the parameters of the CAPs molecular marker (Table 2), the length of the amplicon is designed to be 459 bp, while the T_M_ and GC % are also within the suitable range. There is no complementary binding at the 3′ end of the primer, and the primer does not form a hairpin structure. The electrophoresis bands with the CAP_S_ marker were consistent with the parameters for in silico prediction, which meet the requirements of the subsequent experiments.

### 3.3. Product Detection from PCR Amplifications and Endonuclease Digestion Reactions

Using 56.0 °C as the annealing temperature, the DNA of Nipponbare, WYJ7, R498, ZH11, 9311, NM73, 02428, YX1B, II-32B, and ZH11-SPL14 were used as templates for PCR amplification (Figure 3a). The gel electrophoresis results showed that all amplifications were successful, producing a clear and consistent band for each rice line. After digestion with Hpy188III (Figure 3b), it could be seen that only the PCR product from NM73 was not digested, preserving its single band at 459 bp; whereas, all the other rice lines featured two bands at 279 bp and 180 bp, respectively, indicating digestion by the endonuclease.

Compared to the marker techniques previously used (Table 3), the CAPs marker method developed in this study yields reliable results and requires less amplification time. The sensitivity of the ARMS PCR test is reliant on both the specificity of the primers and the PCR reaction conditions, which include primers’ concentration, annealing temperature, Mg^2+^ concentration, and the Taq DNA polymerase concentration. Incorporating mismatches into dCAP markers may affect amplification efficiency and specificity, leading to decreased reliability and longer development cycles. NGR2 5U S+*Tag* II detected the upstream SNP linkage to *OsGRF4*.

### 3.4. Optimization of LAMP Reaction

We next determined the optimal temperature of the LAMP reaction using NM73 as the DNA template (Figure 4). Figure 4 revealed that the LAMP amplification products formed typical ladder bands. At temperatures 56.0 °C, 56.6 °C, and 57.7 °C, the bands were blurry and faint, indicating reduced amplicons, but were clear and bright at 61.5 °C, 63.2 °C, 64.3 °C, and 65.0 °C, indicating higher concentration of amplicons. Therefore, 65.0 °C was chosen as the optimal temperature for subsequent LAMP reactions.

According to the primer parameters (Table 4), the primary concern is the free energy of the 5′ and 3′ ends, with values of 5′dG and 3′dG all meeting the design requirement of −4 kcal/mol or less. Furthermore, the primer ends were sufficiently stabilized, with the Tm, length, inter-segment distance, and GC content of each region falling within a suitable range for the LAMP reaction to be triggered as expected.

### 3.5. Colorimetric Detection of LAMP Amplification

DNA templates from all 10 different rice cultivars were tested for LAMP amplification and electrophoresis under optimized conditions (Figure 5a). The gel electrophoresis results showed that only NM73 was successfully amplified as a direct result of the precise match of the AA bases of F2/FIP primer to 1187–1188 bp of the *OsGRF4*^NM73^ allele. The colorimetric LAMP reaction was the performed for all 10 rice varieties (Figure 5b). Only one centrifuge tube carrying the NM73 DNA template triggered a colorimetric reaction, turning sky blue in contrast to the other nine tubes, which stayed violet.

Except for NM73, the genotype of the other nine rice varieties was confirmed as *osgrf4*, and is available on public databases (Table 5). Previous research has found that NM73 belongs to the *OsGRF4* genotype [2]. In this work, the endonuclease digestion, LAMP amplification, and colorimetric visual detection results for each rice variety are consistent with the genotype data and satisfy the experimental expectations.

## 4. Discussion

Crop yields increased dramatically during the “Green Revolution” in the 1960s, which was marked by the success of semi-dwarf rice breeding [38]. Semi-dwarf crops have high yield and improved lodging resistance even at high nitrogen inputs, but have lower efficiency of nitrogen uptake from soil as a trade-off [39]. The large amount of nitrogen fertilizer required to achieve the high yield of semi-dwarf crops has resulted in a series of environmental issues such as air pollution, the hole in the ozone layer, climate warming, soil acidification, and water eutrophication [40,41,42]. Excessive nitrogen fertilizer usage will also reduce crops’ resistance to disease and insect pest infestation, making them more vulnerable to infections, which in turn increases the need for pesticides that pose significant food safety issues [43]. As food security is a vital part of the socioeconomic stability of a nation, pursuing crops with high yields and low nitrogen usage are important goals for both academia and relevant authorities [44]. As an example, China’s Ministry of Agriculture had set a target to achieve zero growth rate in the use of chemical fertilizers and pesticides by 2020 [45]. Some researchers have proposed the idea of a new “Green Revolution” [46], centered around the identification and introgression of high NUE genes. In 2018, we discovered that DELLA-GRF4 mediated the coordinated control of crop growth and nitrogen metabolism [2]. The rice growth regulator GRF4 can bind to GIF1 (GRF-INTERACTING FACTOR 1) to activate downstream genes involving nitrogen absorption and assimilation, while DELLA in the GA (gibberellin) pathway inhibits the binding of GRF4 to GIF1. When DELLA accumulates, the interaction between GRF4 and GIF1 is inhibited, causing reduced nitrogen absorption and assimilation. In addition, GRF4 also promotes the expression of genes involved in carbon absorption, which improves rice nitrogen use efficiency by regulating carbon and nitrogen balance. Therefore, it is believed that *OsGRF4*, which promotes various nitrogen-related events is a key gene that could lead to the development of new “Green Revolution” rice varieties with high yield and nitrogen-use efficiency [1].

Grain size is one of the crucial components of rice yield, which has long been the focus of breeding. Currently, numerous genes regulating grain size have been discovered and widely used in breeding, such as *tgw2*, *GS3*, *qTGW3*, *qSW5*, *GW2*, *GS5*, *BG1*, *GW5*, *WTG1*, and *qGL3*, etc. (https://www.ricedata.cn/, accessed on 1 August 2023). Nevertheless, only a few dominant genes, such as *OsGRF4* and *qGL3* [47], are directly responsible for the large grain phenotypes. Breeding of hybrids using these dominant genes is advantageous as the screening and identification process is relatively straightforward, since the F_1_ progenies carrying the dominant gene would display a positive phenotype.

In order to maintain stable rice production as global warming continues, it is crucial to breed new varieties that are more disease and stress resistant. Presently, there is an abundance of genes identified for disease resistance and stress resistance in plants, with some of them being applied extensively in breeding with good results, including *OsGRF4* [48].

As the seed industry becomes more commercialized and marketed, the need for farmers to ensure the purity and authenticity of their rice varieties becomes more urgent [49]. Authentic, high-quality seeds can protect the interests of all parties, including farmers, seed companies, breeders, and consumers, to ensure higher crop yields, increase farmers’ incomes, and maintain the harmonious sustainable development of agriculture. Genotyping the *OsGRF4*^NM73^ gene using the LAMP method is simple, fast, and can be carried out without special equipment. In combination with the rapid preparation of DNA templates [26,50], the visual colorimetric LAMP method can be used by farmers, seed producers, breeders, consumers, and government regulators for the convenient detection of the *OsGRF4*^NM73^ gene and other important genes, to secure the purity and authenticity of rice varieties, and promote the sustainable development of the rice industry.

Therefore, the LAMP method described in this study could serve as a rapid and convenient screening method for rice varieties carrying the *OsGRF4*^NM73^ allele, and, consequently, aids in production of high-yield, high-NUE rice varieties with good disease and stress resistance, while also being useful for the study of other potential functions of the *OsGRF4* allele. While we have achieved our research objective, future study on the incorporation of an additional loop primer pair to our method is warranted to optimize the LAMP detection efficiency in this study.

## 5. Conclusions

Using a literature search and the sequence data available to the public, we successfully designed a CAPs primer pair (*OsGRF4*_2F and *OsGRF4*_2R) that could detect *OsGRF4*^NM73^ in rice with great precision. We then developed a rapid and practical visual detection method for the *OsGRF4*^NM73^ allele in rice using the LAMP technique, which would assist greatly in its application in the new “Green Revolution” rice breeding. Our results also serve as a guide for the development and use of LAMP functional markers for other significant genes.

## Figures and Tables

**Figure 1 genes-14-01850-f001:**
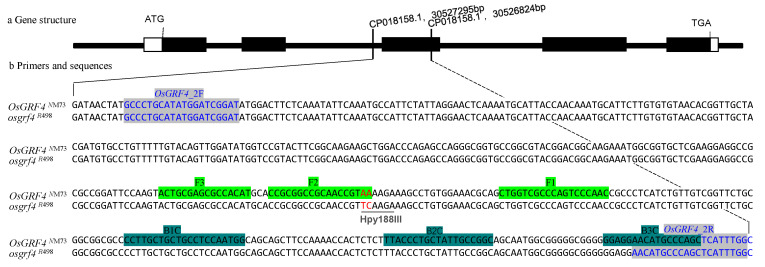
Schematic illustration of primers used for detection of *OsGRF4*^NM73^ genotype in rice. (**a**) Gene structure: The white box before ATG (left) represents the 5′UTR, the white box after TGA (right) represents the 3′UTR, and the black boxes in between represent the exons. The red letters highlight the TC→AA substitution in exon 3 at the location 1187–1188 bp. (**b**) Primers and sequences: *OsGRF4*^NM73^ denotes the *OsGRF4* gene sequence from the rice line NM73, while *osgrf4*^R498^ denotes the sequence from R498. “CP018158.1, 30527295 bp” indicates that the base “G” of accession number CP018158.1 is located at 30527295 bp, whereas “CP018158.1, 30526824 bp” is the same as well. The blue characters beneath the names of the CAPs primers *OsGRF4*_2F and *OsGRF4*_2R indicate the primers’ positions on the gene. The underlined sequence tagged Hpy188III is the recognition site for the restriction enzyme. The placement of the LAMP primers is highlighted (green and teal) underneath labels F3, F2, F1, B1C, B2C, and B3C.

**Figure 2 genes-14-01850-f002:**
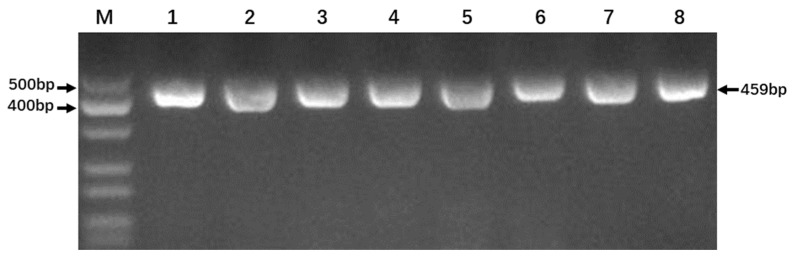
Gel electrophoresis of PCR products produced under different annealing temperatures. Lane M, DNA ladder (MD112, TIANGEN Biotech (Beijing) Co., Ltd., Beijing, China). Lanes 1–8, amplifications at annealing temperatures (Tm) of 56.0 °C, 56.6 °C, 57.7 °C, 59.4 °C, 61.5 °C, 63.2 °C, 64.3 °C, and 65.0 °C, respectively.

**Figure 3 genes-14-01850-f003:**
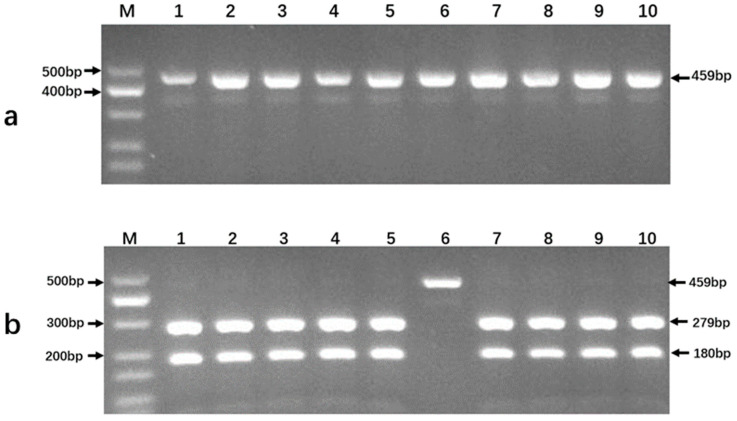
PCR amplification and restriction digestion using DNA templates from different rice lines. (**a**) The PCR amplification results. (**b**) Digestion by Hpy188III endonuclease. Lane M, DNA ladder (MD112, TIANGEN Biotech (Beijing) Co., Ltd., Beijing, China). Lanes 1–10, the rice lines (1. Nipponbare, 2. WYJ7, 3. R498, 4. ZH11, 5. 9311, 6. NM73, 7. 02428, 8. YX1B, 9. II-32B, and 10. ZH11-SPL14).

**Figure 4 genes-14-01850-f004:**
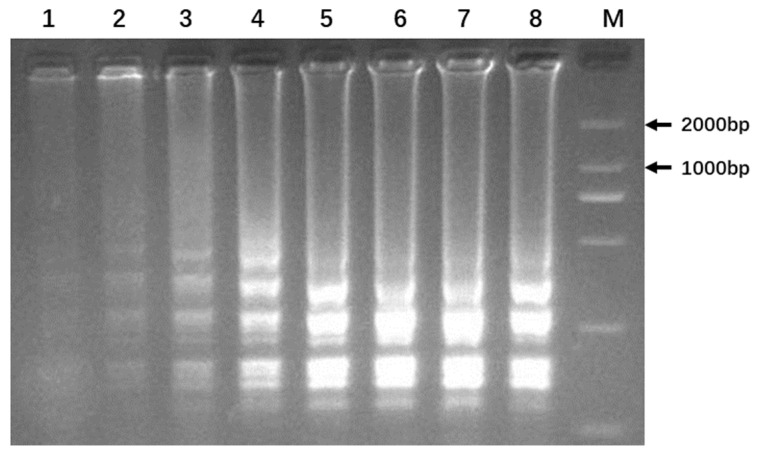
Optimization of LAMP-reaction temperature. Lane M, DNA ladder (MD114, TIANGEN Biotech (Beijing) Co. Ltd., Beijing, China). Lanes 1–8, LAMP reactions at 56.0 °C, 56.6 °C, 57.7 °C, 59.4 °C, 61.5 °C, 63.2 °C, 64.3 °C, and 65.0 °C, respectively.

**Figure 5 genes-14-01850-f005:**
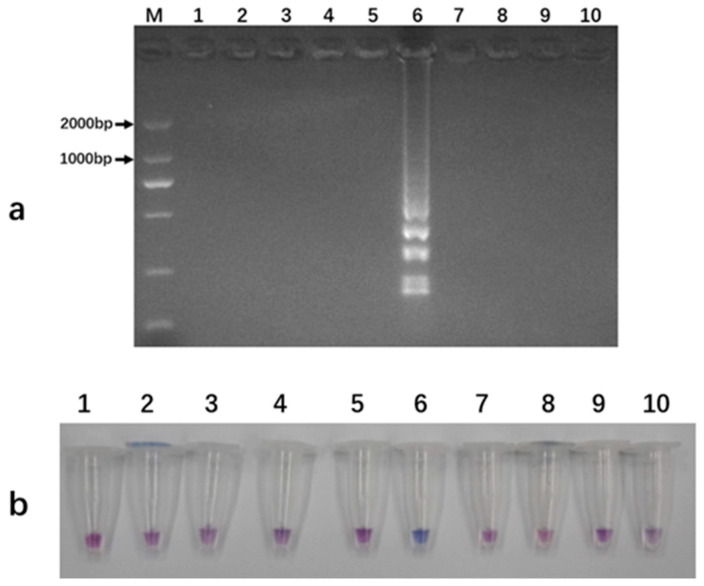
LAMP amplification and colorimetric reaction. (**a**) The universal LAMP amplification. (**b**) Colorimetric LAMP assay. Lane M, DNA ladder (MD114, TIANGEN Biotech (Beijing) Co., Ltd., Beijing, China). Lanes a1–10 and centrifuge tube b1–10: The rice lines (1. Nipponbare, 2. WYJ7, 3. R498, 4. ZH11, 5. 9311, 6. NM73, 7. 02428, 8. YX1B, 9. II-32B, and 10. ZH11-SPL14).

**Table 1 genes-14-01850-t001:** List of different sets of primers designed to detect gene *OsGRF4*^NM73^.

Name	Sequence	Usage
*OsGRF4*_2F	GCCCTGCATATGGATCGGAT	Forward primer for CAPs marker
*OsGRF4*_2R	AATGAGCTGGGCATGTTCCT	Backward primer for CAPs marker
F3	ACTGCGAGCGCCACAT	Forward inner primer for LAMP marker
B3	GCTGGGCATGTTCCTCC	Backward inner primer for LAMP marker
FIP (F1C-F2)	GTTGGGACTGGGCGACCAGCCGCGGCCGCAACCGTAA	Forward outer primer for LAMP marker
BIP (B1C-B2)	CCTTGCTGCTGCCTCCAATGGGCCGGCAATAGCAGGGTAA	Backward outer primer for LAMP marker

**Table 2 genes-14-01850-t002:** In silico prediction of CAPs primer system parameters.

Primer Name	start	end	len	tm	gc%	any_th	3′_th	hp	COMPL	PS
*OsGRF4*_2F	2499	2519	20	59.75	55.00	12.91	0	0	9.33	459
*OsGRF4*_2R	2957	2938	20	59.67	50.00	0	0	0		

Start—Primer 5′ end base position on gene sequence BK004859.1. End—Primer 3′ end base position on gene sequence BK004859.1. Len—primer length (unit, bp). Tm—Primer’s annealing temperatures; PS—PCR amplification product size (unit, bp); any_th—any complementarity TH. 3′_th—3′ End Complementarity TH. Hp—hairpin structure value. COMPL—maximum self-base pairing. Refers to the literature for the algorithm used for calculation for each parameter [32].

**Table 3 genes-14-01850-t003:** Marker methods have been published for the detection of *OsGRF4*.

Marker Name	Type	Seq (5′–3′)	References
PT2	ARMS	AA-R: cgtttccacaggctttctttt	[13]
		TC-R: cgtttccacaggctttcttga	
		PT2-F: ctgtgaaccaacaccctg	
		PT2-R: cggcaatagcagggtaaa	
GS2_SNP	dCAPs	GS2_SNP-F: gagcgccacatgcaccgcggccgcatacgt (*Sna*BI)	[14]
		GS2_SNP-R: ttgcctgttccaccaccaacagc	
NGR2 5U S+*Tag* II	CAPs	F: tcattgacctacggttgc	[15]
		R: gctgctccaacatcttct	

**Table 4 genes-14-01850-t004:** LAMP marker primers and their corresponding parameters.

Primer/Region Name	5′pos	3′pos	len	Tm	5′dG	3′dG	GCrate
F3	2741	2756	16	60.83	−6.57	−5.05	0.63
B3	2952	2936	17	59.23	−6.69	−5.55	0.65
FIP			37				
BIP			40				
F2	2760	2775	18	67.78	−7.87	−5.02	0.72
F1c	2818	2800	19	64.6	−5.61	−5.35	0.68
B2	2916	2898	19	61.88	−7.94	−4.69	0.58
B1c	2853	2873	21	65.96	−5.85	−4.66	0.62

5′pos—the position of base at the 5′ end on gene sequence BK004859.1. 3′pos—the position of base at the 3′ end on gene sequence BK004859.1. Len—primer/region length (bp). Tm—annealing temperatures (°C). 5′dG—the free energy of 5′ end (kcal/mol). 3′dG—the free energy of 5′ end (unit, kcal/mol). GCrate, GC content. The PrimerExplorer V5 software′s user manual contains the algorithms for specific parameters.

**Table 5 genes-14-01850-t005:** Genomic data sources.

Variety Name	Sub-Population	GenBank Assembly Accession	Reference	DOI
Nipponbare	Japonica	GCA_004295705.1		
WYJ7	Japonica		[36]	10.1038/s41422-022-00685-z
R498	Indica	GCA_002151415.1		
ZH11(ZH11-SPL14)	Japonica	GCA_014526345.1		
9311	Indica	GCA_014636015.1		
02428	Japonica		[37]	10.1016/j.cell.2021.04.046
YX1B	Indica		[37]	10.1016/j.cell.2021.04.046
II-32B	Indica		[37]	10.1016/j.cell.2021.04.046

DOI—digital object unique identifier.

## Data Availability

Additional data are available from the corresponding author upon request.

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
