# Peer review of "Rapid Visual Detection of High Nitrogen-Use Efficiency Gene OsGRF4 in Rice (Oryza sativa L.) Using Loop-Mediated Isothermal Amplification Method"

_genes, 2023, doi:10.3390/genes14101850_

Round 1
Reviewer 1 Report
The Authors developed and optimized a simple, low-cost test, useful in small field laboratories for detecting the beneficial allele of the GRF4 transcription factor of cultivated rice varieties. The test allows to distinguish genes that differ only in a two-nucleotide mutation quickly and reliably. The method described should encourage farmers and plant breeders to check the presence of this gene in seed and in rice varieties under selection. If only it is possible to simplify and automate the isolation of genetic material from field samples, the method will be widely used in plant breeding, as well as in agriculture and veterinary medicine, also for mass and cheap identification of pathogens.
Reviewer 2 Report
The reviewed manuscript is dedicated to the design and validation of LAMP-based assay detecting a 2-bp substitution in OsGRF4 gene of rice. The presented results are interesting for scientists, specializing on the field of molecular diagnostics. However, a number of issues needs to be addressed before publication.
Major issues:
1. 3.3. PCR amplification and restriction enzyme digestion, 3.4. LAMP amplification, 3.5 Visual detection of LAMP reaction — reaction volumes, primers concentrations, primers sequences, DNA amount, temperature profiles, HNB concentration and a PCR machine would be highly appreciated.
2. Normally, an additional loop primers pair is designed for LAMP. The pair increases reaction speed and sensitivity.
3. Authors are encouraged to compare the designed LAMP test with previously reported approaches and list possible its limitations. Also, authors mentioned a possibility to test rice samples outside of laboratories. However, no experiments were provided with LAMP and simplified DNA purification procedures, suitable for in-field testing. The latter would greatly increase a manuscript’s value.
Minor issues:
1. Authors are encouraged to add more information about CAPs in an Introduction section.
2. Page 1, line 21: “However, as a rare mutation, a method to.”
3. Page 2, line 58: “detection of OsGRF4 are mostly”
4. Page 2, line 71: “In recent years. LAMP”
5. 2.2. The primer design — GenBank ID for OsGRF4 would help to find the gene sequence.
6. 3.2. Optimization of PCR reaction conditions — what DNA template was used for PCR optimization?
7. Page 5, line 171: “two bands at 259bp and 180bp” — 259 bp instead of 279 bp.
